# Robust point-process Granger causality analysis in presence of exogenous temporal modulations and trial-by-trial variability in spike trains

**Antonino Casile** [1,2]*, **Rose T. Faghih** [3], **Emery N. Brown** [4,5,6]

**1** Istituto Italiano di Tecnologia, Center for Translational Neurophysiology of Speech and Communication (CTNSC), Ferrara, Italy, **2** Harvard Medical School, Department of Neurobiology, Boston, Massachusetts, United States of America, **3** Department of Electrical and Computer Engineering, University of Houston, Houston, Texas, United States of America, **4** Department of Brain and Cognitive Science, Massachusetts Institute of Technology, Cambridge, Massachusetts, United States of America, **5** Picower Institute for Learning and Memory, Massachusetts Institute of Technology, Cambridge, Massachusetts, United States of America, **6** Department of Anesthesia, Critical Care and Pain Medicine, Massachusetts General Hospital, Boston, Massachusetts, United States of America

\* antonino.casile@iit.it, toninocasile@gmail.com

**Data Availability Statement:** The Matlab implementation of our G-ETM and G-ETMV methods is available under a GNU General Public

## Abstract

Assessing directional influences between neurons is instrumental to understand how brain circuits process information. To this end, Granger causality, a technique originally developed for time-continuous signals, has been extended to discrete spike trains. A fundamental assumption of this technique is that the temporal evolution of neuronal responses must be due only to endogenous interactions between recorded units, including self-interactions. This assumption is however rarely met in neurophysiological studies, where the response of each neuron is modulated by other exogenous causes such as, for example, other unobserved units or slow adaptation processes. Here, we propose a novel point-process Granger causality technique that is robust with respect to the two most common exogenous modulations observed in real neuronal responses: within-trial temporal variations in spiking rate and between-trial variability in their magnitudes. This novel method works by explicitly including both types of modulations into the generalized linear model of the neuronal conditional intensity function (CIF). We then assess the causal influence of neuron *i* onto neuron *j* by measuring the relative reduction of neuron *j*'s point process likelihood obtained considering or removing neuron *i*. CIF's hyperparameters are set on a per-neuron basis by minimizing Akaike's information criterion. In synthetic data sets, generated by means of random processes or networks of integrate-and-fire units, the proposed method recovered with high accuracy, sensitivity and robustness the underlying ground-truth connectivity pattern. Application of presently available point-process Granger causality techniques produced instead a significant number of false positive connections. In real spiking responses recorded from neurons in the monkey pre-motor cortex (area F5), our method revealed many causal relationships between neurons as well as the temporal structure of their interactions. Given its robustness our method can be effectively applied to real neuronal data. Furthermore, its explicit estimate of the effects of unobserved causes on

License v3.0 at the following GitHub repository:
https://github.com/toninocasile/G-ETM_G-ETMV.

**Funding:** This work was supported by a grant from the Harvard/MIT Joint Research Program (HM-JRP) to AC and ENB, grant P01-GM118629-02 to ENB, NSF grants #1755780 and #1942585 to RTF and an NEI Core grant (NIH P30EY012196). RTF was partly supported by the HM-JRP grant during her postdoctoral studies. The funders had no role in study design, data collection and analysis, decision to publish, or preparation of the manuscript.

**Competing interests:** The authors have declared that no competing interests exist.

the recorded neuronal firing patterns can help decomposing their temporal variations into endogenous and exogenous components.

## Author summary

Modern techniques in Neuroscience allow to investigate the brain at the network level by studying the functional connectivity between neurons. To this end, Granger causality has been extended to point process spike trains. A fundamental assumption of this technique is that there should be no unobserved causes of temporal variability in the recorded spike trains. This, however, greatly limits its applicability to real neuronal recordings as, very often, not all the sources of variability in neuronal responses can be concurrently recorded. We present here a robust point-process Granger causality technique that overcomes this problem by explicitly incorporating unobserved sources of variability into the model of neuronal spiking responses. In synthetic data sets our new technique correctly recovered the underlying ground-truth functional connectivity between simulated units with a great degree of accuracy. Furthermore, its application to real neuronal recordings revealed many causal relationships between neurons as well as the temporal structure of their interactions. Our results suggest that our novel Granger causality method is robust and it can be used to study the function connectivity between a set of simultaneously recorded spiking neurons, even in presence of unobserved causes of temporal modulations.

## Introduction

Modern neurophysiological recording techniques allow to simultaneously probe the activities of tens to hundreds of neurons [1–3]. The availability of these high-dimensional data sets allows to address novel and relevant research questions about the brain. A particularly important question is to investigate brain functions at the circuit level, by assessing the influences between neurons. To this end, several analytical tools have been proposed in the past, such as cross-correlogram [4], joint peri-stimulus histogram [5] or gravitational cluster [6]. While providing noteworthy insights, these tools have also limitations as (1) they do provide little information about the directionality of discovered interactions and (2) they do not usually consider the point-process nature of neuronal spike trains. To overcome both issues Kim et al. proposed an extension of Granger causality to point processes [7].

   Granger causality is an analytical tool originally proposed in the context of econometric time series [8]. A stochastic process $x$ is said to Granger causally influence another process $y$ (henceforth denoted with $x \rightarrow y$) if knowledge of values of $x$ at times before $t$ improves, in a statistically significant manner, the prediction of $y$ at time $t$ beyond inclusion of past values of $y$ itself. Granger causality assumes that all sources of temporal modulations of the processes $x$ and $y$ must be endogenous to the set of considered processes. That is, they must be entirely explained by the processes' past histories and there should be no common unobserved driver of temporal variability [9]. However, this is often not the case in neurophysiological experiments, where many of the causes that produce temporal modulations in neuronal responses are exogenous to the ensemble of recorded neurons. Indeed, the activity of a neuron at each time point results from the integration of signals coming from many, potentially thousands, other neurons, most of which are not concurrently recorded. Furthermore, in many experimental settings, we are interested in the so-called *functional connectivity* between neurons.

That is, the amount and directionality of influences between neurons when the brain changes its state as a consequence of, for example, sensory stimulation or motor behavior. Under these conditions, neurons exhibit temporal modulations in the statistics of their firing patterns that are due to the interactions with neighboring neurons located in the their local network as well as more distant units in projecting brain regions. Finally, the magnitude of neuronal responses often exhibits a physiological, potentially correlated, trial-by-trial variability, that brings the system further away from the conditions assumed by Granger causality.

In this paper, we show that, in presence of exogenously temporally modulated and trial-by-trial variable spike trains the point-process Granger causality technique proposed by Kim et al. [7] might recover inaccurate patterns of connectivity. We then propose two novel methods that address this issue. The first method, called G-ETM (Granger causality with Exogenous Temporal Modulations), is designed to extend point-process Granger causality to spike trains whose magnitudes are modulated by exogenous, unobserved, causes. The second method, called G-ETMV (Granger causality with Exogenous Temporal Modulations and trial-by-trial Variability), is computationally more demanding and it recovers the correct pattens of functional connectivity between a set of interconnected neurons exhibiting both trial-by-trial variability and exogenous temporal modulations in their firing patterns. Both methods work by adding covariates to Kim et al.'s Granger causality model. In G-ETM, these covariates model instantaneous changes in firing probability that cannot be attributed to the neurons' past history. In G-ETMV, they provide a factor that scales neurons' firing rate on a trial-by-trial basis so as to account for trial-wide modulations of neuronal activities. We show the effectiveness of our new Granger causality techniques by means of quantitative computer simulations and application to real spike trains recorded from the monkey pre-motor cortex (area F5).

## Results

Throughout this section we will denote temporal modulations in neuronal responses that are due to interactions between the recorded neurons (including self-interactions) as *endogenous* and temporal modulations that are due to unobserved causes as *exogenous*.

### Standard point-process Granger causality fails with spike trains exhibiting exogenous temporal modulations

In Kim et al.'s original point-process Granger causality method the conditional intensity function (CIF) $\lambda_i$ of a neuron $i$ is modeled as the product of a baseline firing rate $\gamma_{i,0}$ and a factor that depends, through the to-be-estimated parameters $\gamma$, on the past histories of all neurons in the ensemble, including neuron $i$ itself (Eq 3 in Table 1 and in the Methods sections). To show how such model can produce incorrect patterns of connectivity in the presence of spike trains exhibiting exogenous temporal modulations, we applied Kim et al.'s Granger method to 40

**Table 1. Eq. 3 represents the point-process Granger causality model proposed by Kim et al. [7].** Eqs. 5 and 7 represent the G-ETM and G-ETMV models proposed here respectively.

| Method | Mathematical Model |
|---|---|
| Kim et al. | $$log\lambda_i(t|\gamma_i, H_i(t)) = \gamma_{i,0} + \sum_{q=1}^{Q}\sum_{m=1}^{M_i}\gamma_{i,q,m}R_{q,m}(t) \qquad (3)$$ |
| G-ETM | $$log\lambda_i(t|\gamma_i, H_i(t)) = \alpha_{i,c_i(t)} + \sum_{q=1}^{Q}\sum_{m=1}^{M_i}\gamma_{i,q,m}R_{q,m}(t) \qquad (5)$$ |
| G-ETMV | $$log\lambda_{i,p}(t|\gamma_i, H_i(t)) = \beta_{i,p} + \alpha_{i,c_i(t)} + \sum_{q=1}^{Q}\sum_{m=1}^{M_i}\gamma_{i,q,m}R_{q,m}(t) \qquad (7)$$ |

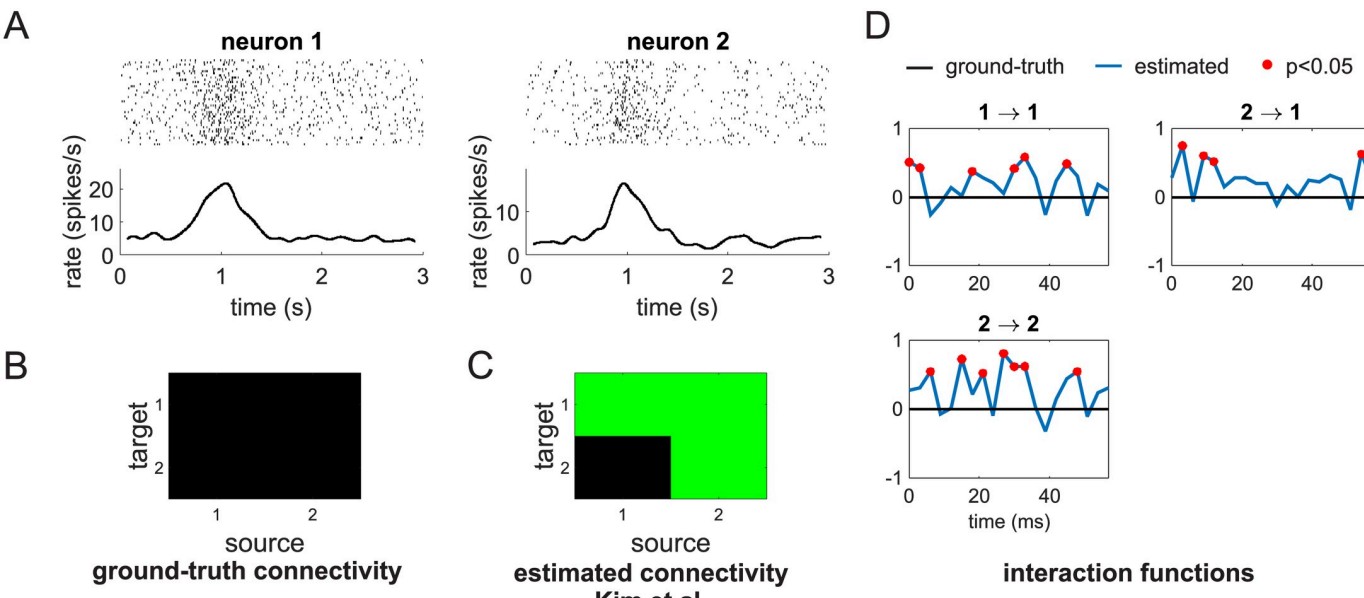

**Fig 1. Standard Granger causality fails with spike trains exhibiting exogenous temporal modulations.** (A) Spike trains of two units simulated by means of two independent Poisson processes. In the top panels, each row represent a trial (a total of 40 trials were generated) and each vertical line a spike. The bottom panels show the average firing rate across trials. On each trial, each neuron underwent a bell-shaped modulation of its firing rate centered around $t = 1\ s$ and with a temporal width of .2 $s$. (B) Ground-truth connectivity of the two units. In this representation a green square represents a functional connection from the source to the target unit, while a black square signifies no connection between them. Since the two units are independent the ground-truth connectivity matrix contains, in this case, only black squares. (C) Connectivity recovered by the point process Granger causality technique proposed by Kim et al. [7]. The recovered connectivity matrix contains three fictitious connections: $1 \rightarrow 1$, $1 \rightarrow 2$ and $2 \rightarrow 1$. (D) Ground-truth values (black curves) and estimates (blue curves) of the interaction functions for the significant functional connections. Red dots mark values that are significantly different from 0 at $p < 0.05$.

simulated trials (Fig 1A) of a simple system consisting of two units. The two units were not functionally connected as their spike trains were generated by means of two independent Poisson processes (Fig 1B) and their firing rates underwent an exogenous bell-shaped modulation centered around $t = 1\ s$. Responses like these might be recorded, for example, in motor areas during the execution or preparation of actions (see for Example Fig 2 in [10]). They beg the obvious question of whether the two units represent subsequent stages of cortical processing, and their responses are thus causally related, or if they are independently driven by an external, unobserved source.

This relevant question represents a natural application of the Granger causality framework. Application of Kim et al.'s method to the spikes trains in Fig 1A revealed many causal connections that, although statistically significant, were not actually present in our system (compare the ground-truth connectivity in Fig 1B with the recovered connectivity in Fig 1C). To see why this happened we have to consider the estimates of the interaction functions (the $\gamma$ terms in Eq 3 and in Fig 1D). In the Granger framework, interaction functions model how the past history of all neurons at different time lags modulate, at each time point, the activity of a given neuron $i$. In our example, their ground-truth values are identically zero for all neurons and time lags as there is no mutual or self interaction at any time lag between the two simulated units. However, not only their estimated values are different from zero at several time lags, but, in many cases, these differences are also statistically significant (red dots in Fig 1D). We can explain these results both at the theoretical and at the implementation level.

At a theoretical level, in the Granger-causality framework, events $B$ are said to be the "cause" of another event $A$ if (1) they precede $A$ in time, (2) knowledge of $B$ change our uncertainty about $A$ beyond inclusion of all other available information. This can be expressed, in

mathematical notation, in terms of conditional probabilities as:

$$P(A) \neq P(A|B) = P(A|B, C, D) \qquad \forall \ C, D \tag{1}$$

where $P(A)$ is a probability distribution representing our knowledge of the events $A$ and $P(A|\mathbb{D})$ is the probability of $A$ conditioned on a set $\mathbb{D}$ of events and $C$ and $D$ are events that are supposed to be irrelevant. If events not included in $B$ turn out to be relevant, say $C$ in Eq 1, then this might lead to spurious causality links [8]. Indeed, in this case, we would have:

$$P(A) \neq P(A|B) \neq P(A|B, C) = P(A|B, C, D) \qquad \forall \ D$$

which violates the basic assumption of Granger causality (i.e. Eq 1). In Kim et al.'s Granger causality method, the exogenous modulations of neuronal spike trains are not included in the set $B$ of predictors [7] and its application to such data sets becomes thus meaningless.

At the implementation level, the GLM fitting process assigned the variance produced by the temporal modulations of the spike trains to the only available free parameters, which in Kim et al.'s original Granger causality method, are those related to interactions between neurons (the $\gamma$ terms in Eq 3). For the specific data set of Fig 1A, inclusion of fictitious causal influences $1 \rightarrow 1$, $2 \rightarrow 2$ and $2 \rightarrow 1$ could indeed explain a significant fraction of the variance contained in the spike trains. This result is not only incorrect but also not robust. Different data sets, generated according to the same CIFs as those in Fig 1, will, in general, produce different fictitious patterns of causal connectivity.

## Extending point-process Granger causality to spike trains exhibiting exogenous temporal modulations

To overcome this problem we propose here G-ETM (Granger causality with Exogenous Temporal Modulations): a novel model that extends the computation of Granger causality to spike trains exhibiting exogenous temporal modulations. To this end, we exploited the organization of neurophysiological experiments into trials and the consistency, across trials, of temporal changes in firing rates to divide, for each neuron $i$, the duration $T$ of each trial into $N_i$ non-overlapping windows. Within each window, we model the CIF of a given neuron $i$ as the sum of a *baseline* rate of activity (the $\alpha$ terms in Eq 5) and the sum of the influences of all other neurons in the ensemble, including neuron $i$ itself (the $\gamma$ terms in Eq 5). Having one additional parameter for each interval allows us to explicitly take into account transient changes in the CIF of neurons due to unobserved factors that cannot be estimated based on their past histories.

Application of G-ETM to the spike trains of Fig 1 produced the correct pattern of causal connectivity (Fig 2A). Furthermore, our technique produced also an estimate of the exogenous temporal modulations of the two simulated units that correctly captured their ground-truth values (Fig 2B). This happened because we now explicitly model exogenous temporal changes of firing rates by means of the parameters $\alpha$ in Eq 5. Therefore, the GLM fitting process no longer needs to generate fictitious connections to explain the variance that they produce.

We next compared G-ETM and Kim et al.'s model on a more complex system composed of 9 units subdivided into two disjoint (i.e. not interacting) subsets: units 1-3 and 4-9 (Fig 3A) respectively. Within each simulated 3 s trial, units' firing patterns were determined by (1) influences from other units in the same subset and (2) bell-like exogenous stimulation that for each unit peaked at a different time in the interval between $t = 1$ $s$ and $t = 2$ $s$. This example is meant to model the case of simultaneous recordings from two areas during occurrence of an experimental event. In this setting, the question arises of whether there is any functional connectivity between the two recorded areas and, if so, what is its directionality. In our simulated

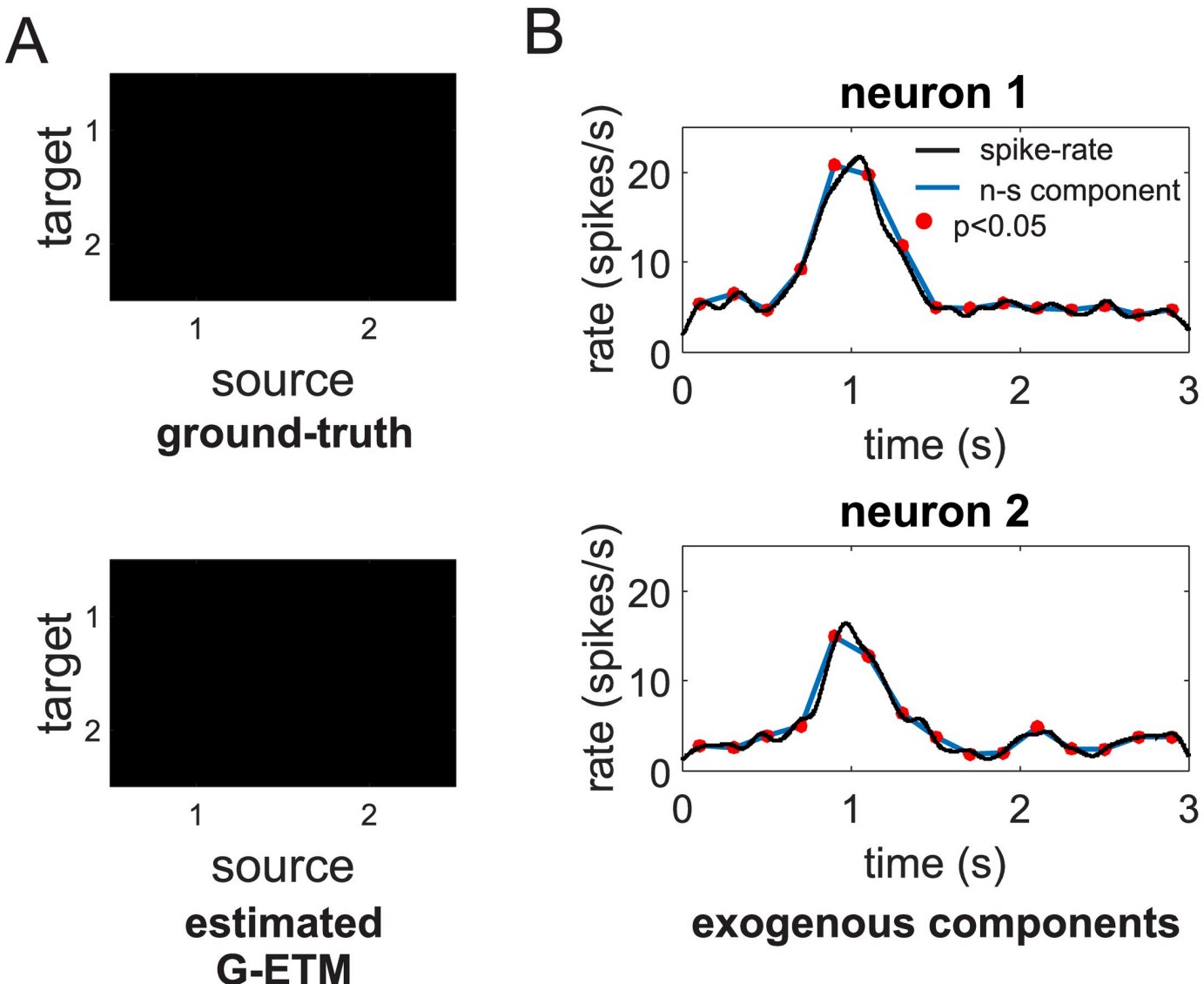

**Fig 2. Application of our G-ETM Granger causality method to the spike trains of Fig 1.** (A) ground-truth (upper panel) and recovered (bottom panel) connectivity matrices. (B) Estimates of the exogenous components of firing patterns. That is, changes in firing rates that are not due to interactions with other neurons. Symbols are as in Fig 1.

network, there was no direct connectivity between the two *areas* (i.e. the two subsets of units). Application of Kim et al.'s method provided an inaccurate estimate of the local pattern of the connectivity both within and between the two subsets of units (Fig 3B). In particular, it produced several additional false-positive connections suggesting an incorrect pattern of *inter-area* connectivity. In an experimental setting, this pattern of results would provide support for the incorrect conclusion of a functional connectivity between the two *areas*. On the contrary, G-ETM recovered the correct pattern of causal connectivity both within and between the two subsets of units (Fig 3C). Furthermore, it also provided an accurate estimate of the interaction functions between units (Fig 3D). It is worth noting that temporal changes in the units' firing rates were almost entirely due to exogenous stimulation (Fig 3E). This means that our method

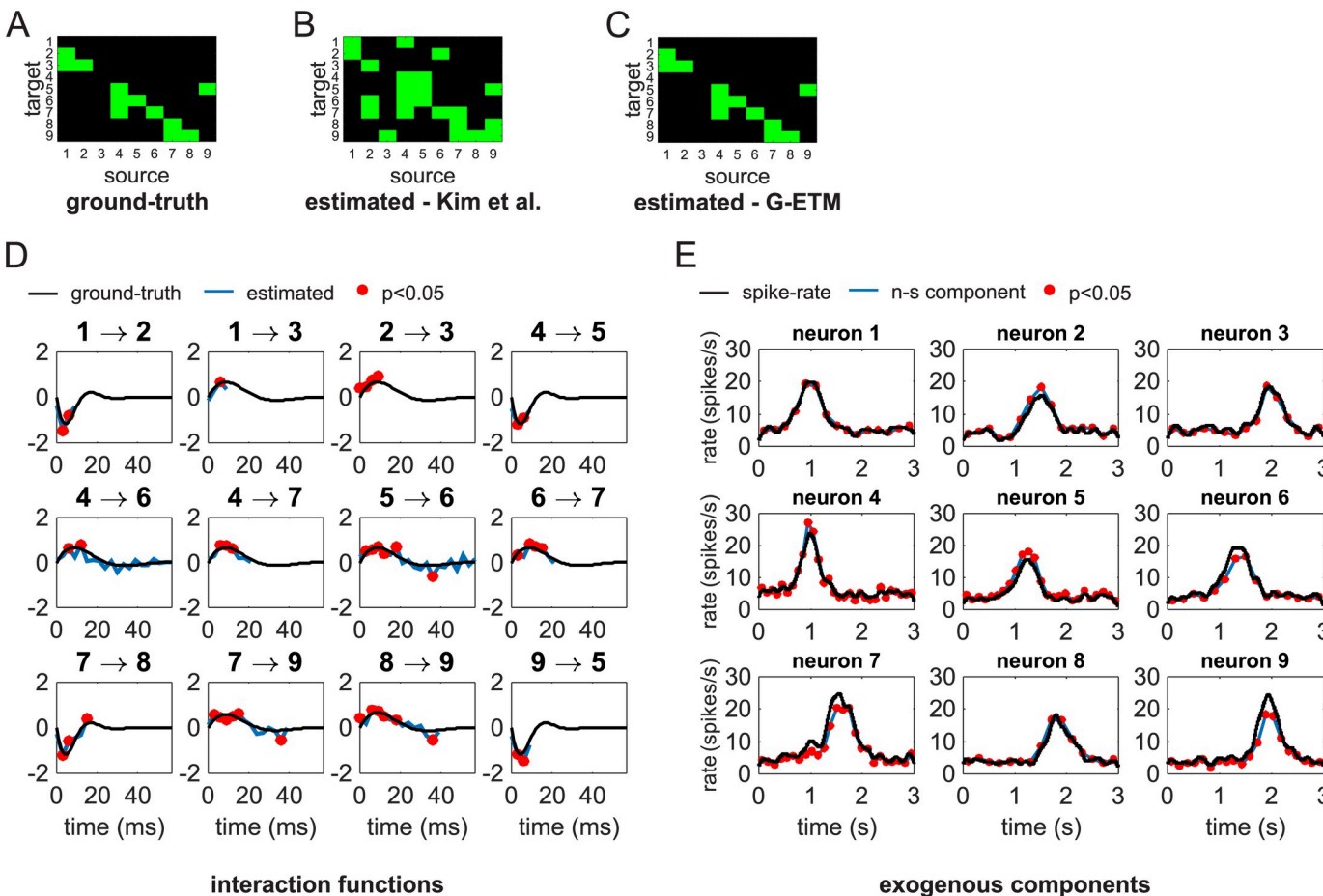

**Fig 3. Application of G-ETM to a complex system.** (A) Ground-truth connectivity pattern. (B) Connectivity pattern estimated by applying Kim et al.'s method [7]. (C) Connectivity pattern estimated by means of our G-ETM method. (D) Estimated (blue curves) and ground-truth (black curves) interaction functions of the significant causal connections. (E) Estimates of the exogenous components of firing patterns. Symbols are as in Figs 1 and 2.

was sensitive enough to detect influences between units, even when, as it is often the case for real neurons, they produced only minimal changes in their firing rates.

To provide a more thorough comparison of G-ETM and Kim et al.' method we performed a series of Monte Carlo simulations (Fig 4). To this end, we simulated 40 trials of a network consisting of 4 neurons and 6 connections whose placement (i.e. connected nodes and directionality of the connection), type (i.e. excitatory or inhibitory) and strength were randomly determined (but, of course, it did not change across trials). In addition to mutual and self influences the spike rates of the 4 neurons underwent also an exogenous bell-shaped modulation. For each neuron, the modulation peaked always at the same time that was however different across neurons and uniformly distributed in the interval $t = 1\ s$ until $t = 2\ s$. We then estimated causal connectivity by applying both Kim et al.'s and our method and compared these two connectivity patterns with their known ground-truth values (Fig 4A). We computed the percentage of correct responses by dividing the number of correctly detected functional connections by 6 (the ground-truth value of functional connections present in the network) and the percentage of false positive by dividing the number of incorrectly reported functional connections by 10 (the ground-truth value of non-connected node pairs). We iterated this procedure 100 times. At each run, we randomly set the network structure and computed the percentage

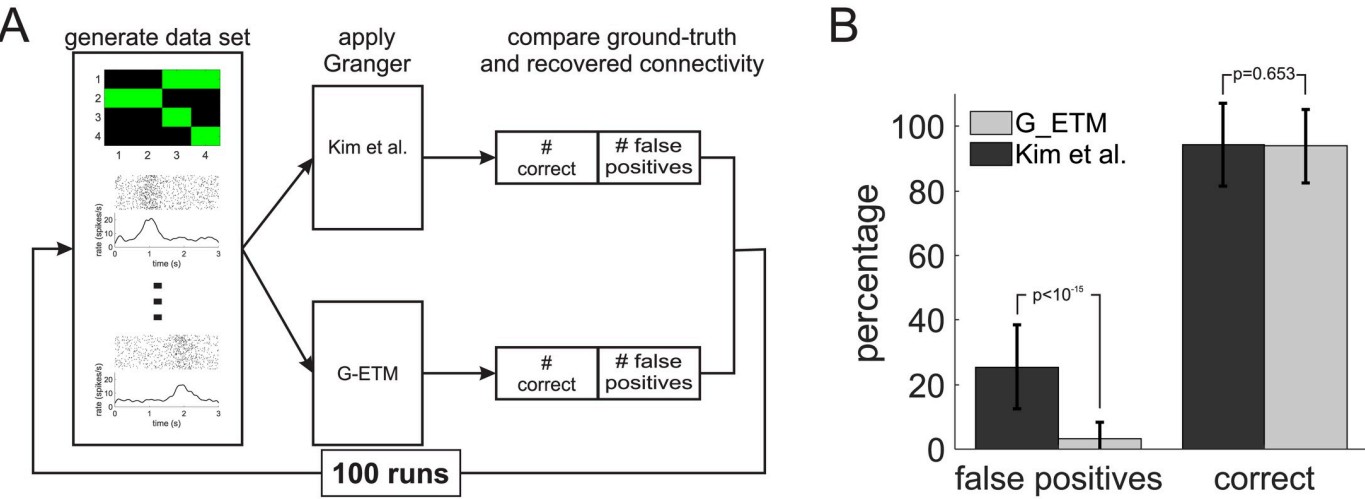

**Fig 4. Monte Carlo comparison of G-ETM and Kim et al.'s model.** (A) Pictorial exemplification of our procedure (see main text for further details). In brief, we first randomly generated a connectivity pattern in a network of 4 neurons. We then applied G-ETM and Kim et al.'s Granger techniques to a data set consisting of 40 simulated trials for each neuron. Finally, we compared ground-truth connectivity with that estimated by the two methods. (B) We repeated this procedure for 100 runs to estimate the percentage of correct and false positive connections recovered by the two methods. Error bars represent standard deviations. The numbers above vertical bars signify probability.

of correct responses and false positives produced by G-ETM and Kim et al.'s method respectively.

Consistent with the intuition provided by Figs 1 and 3 application of Kim et al's method produced false positives (i.e. deeming a connection significant when it is not present in the network) in 26% of the cases, which exceeds by 5-fold the set statistical threshold of $p < 0.05$ (Fig 4B). On the contrary, G-ETM not only provided a comparably good estimate of the connectivity pattern (approximately 94% for both methods) but also produced a percentage of false positives (3.2%, Fig 4B) that was compatible with the selected statistical threshold ($p < 0.05$).

The spike trains used to generate the results in Fig 4 were obtained by a set of point-process random processes that matched the assumptions of our model. While this is necessary to validate G-ETM's implementation, it is also important to explore how our method behaves in presence of spike trains that are generated by processes potentially violating its assumptions. To address this question we generated spike trains by means of a network of integrate-and-fire units (see Methods section for further details). In such network, the spike generation process departs in two important ways from G-ETM's assumptions. In our G-ETM method, functional influences are assumed to modulate the target neuron's firing rate (1) directly and (2) in a multiplicative manner (the double summation in Eq 5). On the contrary, in the integrate-and-fire network that we used to generate synthetic spike trains, source neurons influence a target neuron's firing rate only (1) indirectly through its membrane potential and they do so in (2) an additive manner. That is, rather than having a multiplicative effect on the target neuron's firing rate, a spike from neuron *i* functionally influences another neuron *j* by producing an increase (excitatory influence) or decrease (inhibitory influence) in neuron *j*'s membrane potential [11], which is independent from its instantaneous value. We investigated G-ETM's robustness to these violations of its assumptions in a set of additional Monte Carlo simulations, which followed the same procedure of Fig 4A with the only notable exception that spike trains were generated by means of a network of 4 integrate-and-fire units. Results in Fig 5 show that, despite these major departures from our model's assumptions, G-ETM still provided a very good

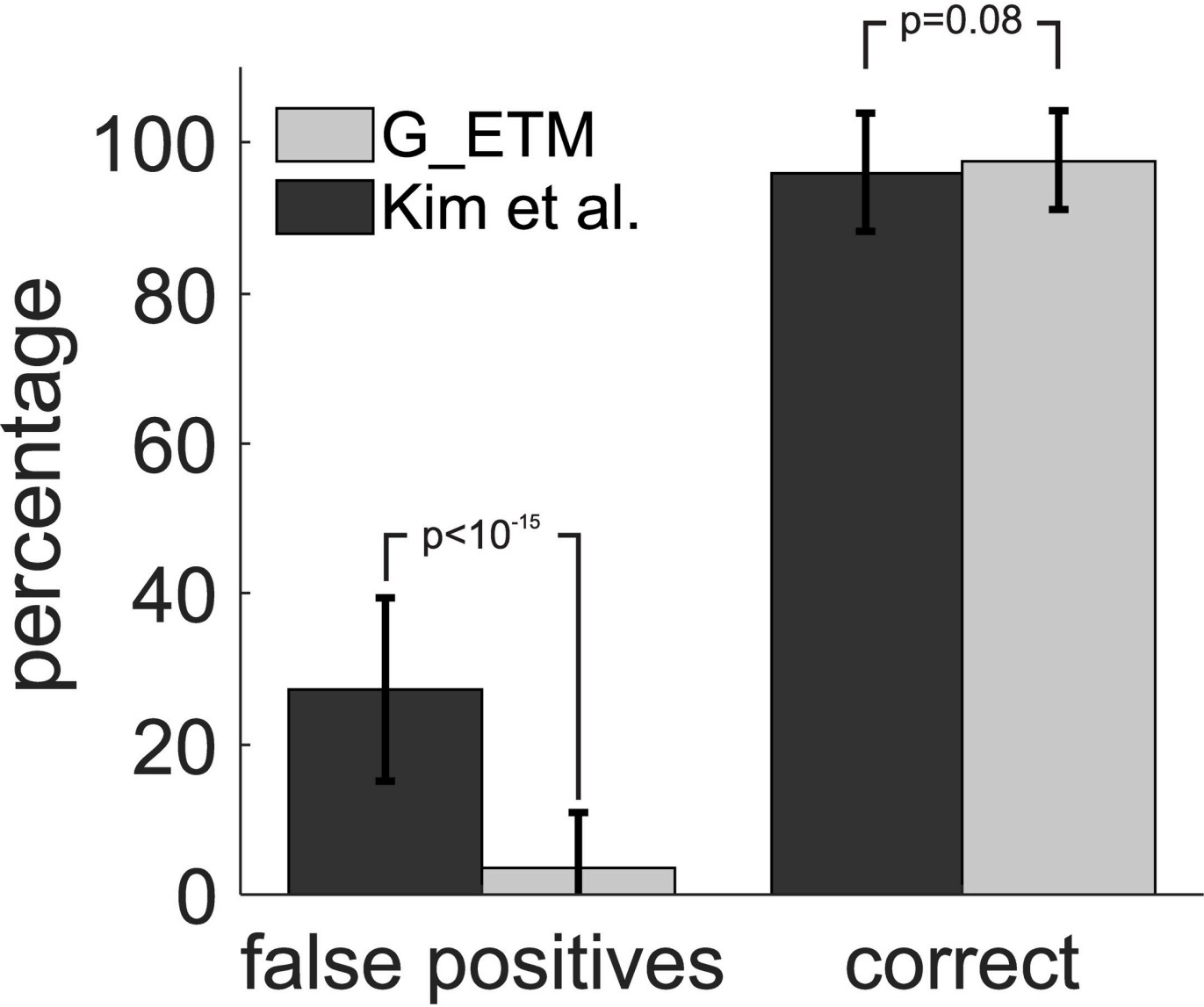

**Fig 5. Monte Carlo comparison of G-ETM and Kim et al.'s model on data generated by means of integrate-and-fire units.** Vertical bars represent the results of Monte Carlo simulations following the same procedure as in Fig 4A, with the only notable exception that spike trains were generated by means of a network of 4 integrate-and-fire units. Statistics are computed as in Fig 4. Error bars represent standard deviations. The numbers above vertical bars signify probability.

estimate of the connectivity patterns. Indeed, it recovered functional connections with high accuracy (98%). Notably, the percentage of false positives (3.6%) was compatible with the set statistical threshold of $p < 0.05$. On the contrary, application of Kim et al's method produced, similar to the results shown in Fig 4, a percentage of false positives (27.3%) that dramatically exceeded the set statistical threshold.

Taken together, the results of Figs 4 and 5 show that G-ETM provides an accurate estimate of the causal influences in a network of neurons in the presence of exogenous temporal modulations of their firing rates. Notably, G-ETM's performance were robust to violations of its underlying assumptions.

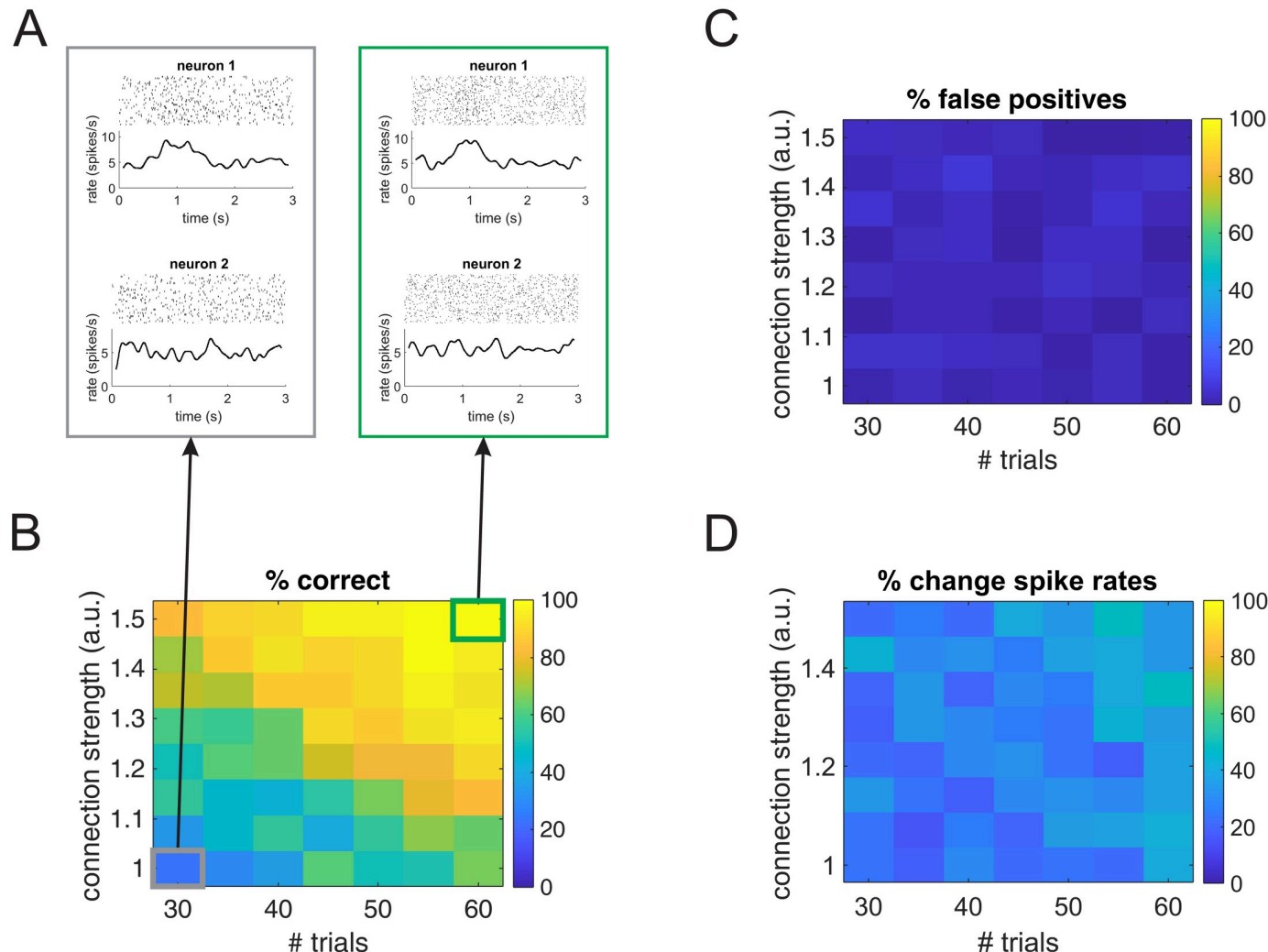

**Fig 6. Sensitivity of G-ETM.** The four panels show the results of the application of G-ETM to a simple network consisting of two neurons with a directed functional connection from neuron 1 to neuron 2 when the strength of the functional connection and number of available trials were parametrically changed. (A) Examples of spike trains generated in the case of 40 trials and a strength of 1 (gray box) and 60 trials and a strength of 1.5 (green box). (B, C) Percentage of runs in which G-ETM correctly detected the functional connection from neuron 1 to neuron 2 (panel B) or erroneously reported a false positive (panel C) as a function of the strength of the functional connection and number of trials. The gray and green highlighted conditions correspond to the two example data sets shown in Panel A. (D) Percentage of trials in which the firing rate of neuron 1 in the interval $[.8, 1.2]s$ was larger than in the interval $[1.8, 2.2]s$.

## Sensitivity of G-ETM

A question that arises when analyzing a method for investigating functional connectivity is that of its sensitivity. That is, how likely is the proposed method to detect a functional connection when it is truly there. In our G-ETM method, two factors that, among others, play an important role in detecting a functional connection between the activities of two units are (1) the strength of the directed influence and (2) the number of available trials. We quantitatively investigated the role of these two factors in a further set of Monte Carlo simulations. To keep our simulations computationally tractable, we focused on a simple network consisting of two neurons with a functional connection $1 \rightarrow 2$ from neuron 1 to neuron 2 and in which the spike rate of neuron 1 exhibits a Gaussian-shaped increment centered around $t = 1s$ and with a variance of $0.2s$ (Fig 6A). This simple model is meant to represent the case of two units

belonging to two networks/areas respectively exhibiting a directed functional connection from one onto the other.

To investigate G-ETM's sensitivity, we parametrically changed the strength of the functional connection of neuron 1 onto neuron 2 and the number of generated trials. For each combination of strength of the functional connection and number of trials, we performed 50 runs. At each run we generated a new set of spike trains and applied our G-ETM method. Fig 6B and 6C show the percentages of correct responses (Fig 6B) and false positives (Fig 6C) detected by G-ETM across runs, as a function of the connection strength and number of available trials. These results suggest that G-ETM's sensitivity is modulated in a similar manner by the strength of the functional connection and the number of available trials, at least in the explored region of the parameter space. That is, G-ETMwas more likely to detect a weaker functional connection when more trials were provided. Conversely, fewer trials were necessary to detect a stronger functional connection. Crucially, the percentage of false positives remained compatible with the set statistical threshold of $p < 0.05$ across the explored range of parameters' values (the grand average of false positives was 2.7%).

To fully gauge the significance of the results in Fig 6B and 6C we developed a simple benchmark to compare them against. To this end, let's notice that, given the structure of our network, a spike emitted by neuron 1 increases the likelihood of neuron 2 to fire. Thus, an alternative method to detect whether neuron 1 is influencing neuron 2 is to measure if neuron 2's firing rate increases in the period around $t = 1$ of maximum activation of neuron 1 compared to baseline conditions. We performed such a measure, by testing, across trials, whether the firing rate of neuron 2 in the interval $[.8, 1.2)s$ was significantly larger than in the interval $[1.8, 2.2)s$ (i.e. baseline condition). It must be emphasized that devising such a test needed the full knowledge of the network (i.e. it needed the knowledge that neuron 1 is projecting onto neuron 2 and the latter receives no other input) and yet it was a poor estimator of the functional influence on one neuron onto another (Fig 6D). Indeed, the percentage of runs in which this test was significant was consistently smaller than the performance of our G-ETM method, with, notably, no obvious dependence on either the connection strength or the number of trials (comare Fig 6B and 6D). Taken together, the results of Fig 6 show that, in a large region of the investigated parameter space, our G-ETM method can detect functional influences between neurons that are so weak as to not appreciably modulate the target neuron's firing rate.

## Application to real spike-train data

In a further step we applied G-ETM to real spike-train data. To this end, we simultaneously recorded the response of 12 neurons from the monkey pre-motor cortex (area F5) during the preparation of goal-directed motor acts. The task of the monkey was to attend to a briefly flashed cue indicating a to-be-executed action and to withhold movement execution until a subsequent go signal occurring randomly in the time interval comprised between 0.8 and 1.2s after cue onset. S1 Fig shows the responses of all 12 recorded neurons during the motor preparation period. In each panel, $t = 0$ marks cue presentation.

We collected data from a total of 57 trials and analyzed neuronal responses recorded in the interval from 0.5 s before until 1 s after cue presentation. Consistent with previous studies of monkey pre-motor cortex [12], the responses of neurons in area F5 were significantly modulated by the preparation of a motor act, exhibiting both phasic and transient modulations in their firing rates (S1 Fig). We applied G-ETM to these spike trains. The results of our analysis revealed a complex pattern of Granger connectivity with both self- and mutual interactions between the recorded neurons (Fig 7A). Application of Kim et al.'s method recovered a

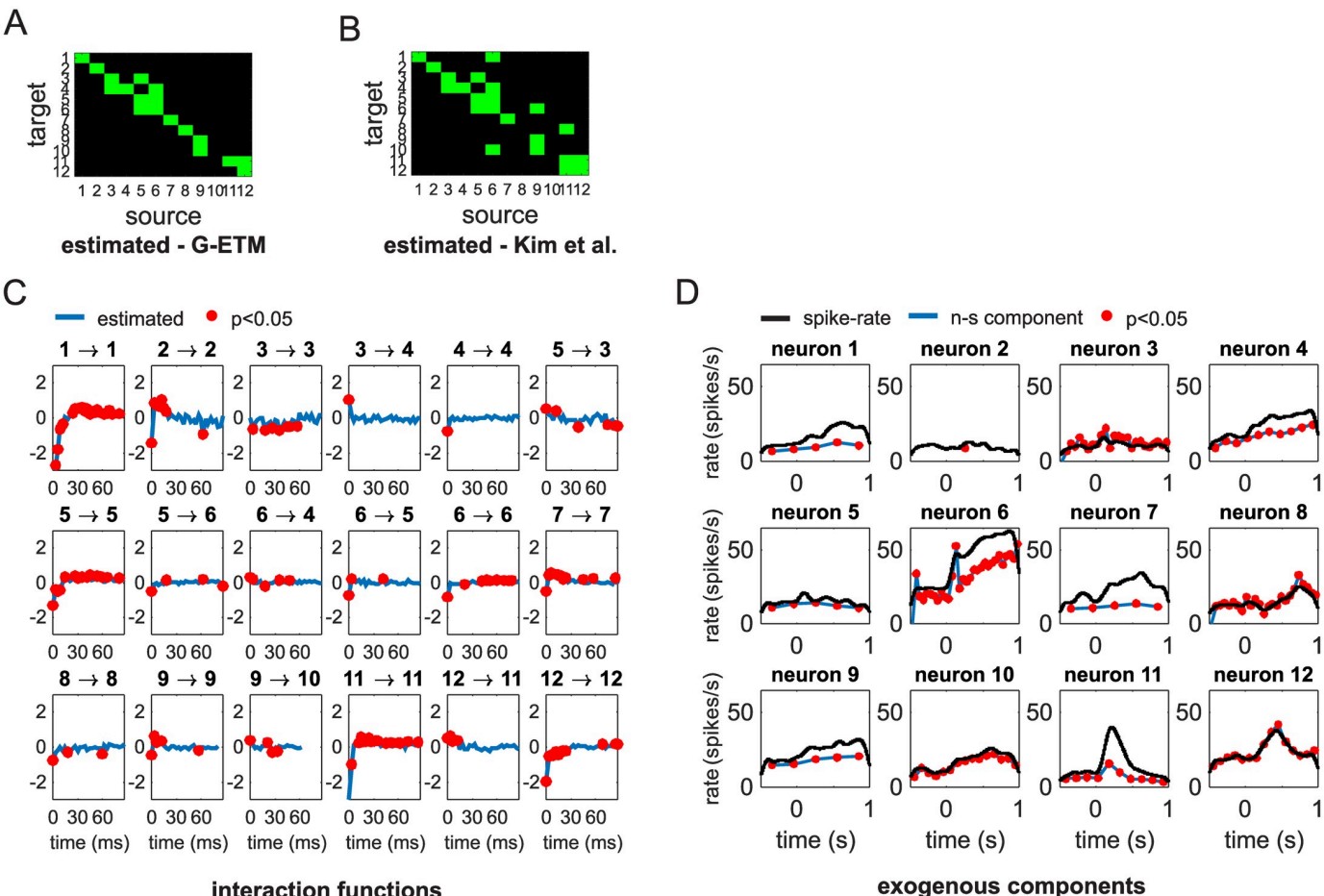

**Fig 7. Application of G-ETM to real spike trains.** (A,B) Connectivity pattern estimated by our G-ETM (panel A) Kim et al.'s (panel B). (C) Estimated interaction functions of the significant causal connections. (D) Estimates of the exogenous components of firing patterns. Symbols are as in Fig 3.

different pattern of connectivity exhibiting a higher number of mutual influences between neurons (off-diagonal elements in Fig 7B). From results shown in Figs 3 and 4, we know that G-ETM is more likely to recover the correct pattern of connectivity when neuronal activations exhibit exogenous modulations. Since this is the case with the analyzed spike trains (see S1 Fig) then we should regard the results provided by our G-ETM model as more likely to be correct.

Interestingly, examination of the recovered interaction functions (Fig 7C) suggests that both self- and mutual interactions are time-dependent with a general trend of being inhibitory at shorter time scales and excitatory at longer time scales. This temporal dynamics is compatible with a scenario in which source neurons cause an increase in firing probability of target neurons (late excitatory component) after an initial transmission delay of their influence (early inhibitory component). The extent of these delays can be estimated based on the time at which the curves in Fig 7C cross the x-axis. Finally, examination of the recovered exogenous components of the firing patterns (Fig 7D) shows that for some units (e.g. units 6 or 11) their temporal modulations could be only partially explained by exogenous influences and the remaining part could be explained by self- or mutual interactions with other units. This result suggests that, in addition to recovering patterns of causal connectivity, G-ETM can be also effectively

used to decompose the firing pattern of recorded units into exogenous (i.e. due to unobserved units/causes) and endogenous (i.e. due to observed units/causes) components.

## Accounting for trial-by-trial variability

We have so far assumed that the stimulus-evoked responses of neurons are stereotyped and do not change across trials. However, while maintaining the same overall *shape*, the magnitude of neuronal firing patterns can often exhibit considerable variability across trials. It has been shown that these trial-by-trial variations can produce spurious patterns of Granger causality and this problem becomes even more severe when these variations are correlated across neurons [13, 14]. Fig 8 shows an example of such problems in a very simple system composed of two simulated units. In this example, on each trial $p$, the activity of unit $i$ was generated by means of an inhomogeneous Poisson process with firing probability $A_{i,p} \cdot \lambda_i(t)$, where the factor $A_{i,p}$ sets the overall magnitude of the response $\lambda_i(t)$ in trial $p$. The processes $\lambda_1$ and $\lambda_2$ were independent and both underwent a bell-shaped temporal modulation of their firing rates centered at $t = 1$ (Fig 8B). We set $A_{1,p} = A_{2,p}, \forall p$ to correlate the trial-by-trial variability of the two units (Fig 8C). Application of G-ETM recovered in this case an incorrect pattern of causal connectivity. This happened because trial-by-trial changes in response magnitude produced additional variance in the data that could not be accounted for by the exogenous components of our G-ETM model (see the mismatch between the blue and black curves in Fig 8E). Therefore, the GLM fitting process attempted to explain this additional variance by means of the other available free parameters, which are those related to interactions between neurons. Indeed, for this specific realization of spike trains, inclusion of fictitious causal influences $1 \rightarrow 2$, $2 \rightarrow 1$ and $2 \rightarrow 2$ significantly improved the percentage of explained variance (Fig 8F) thus producing an incorrect estimate of the pattern of functional connectivity.

To take into account correlated trial-by-trial variability in the magnitude of neuronal responses we extended our G-ETM model. To this end, we further augmented it with a set of

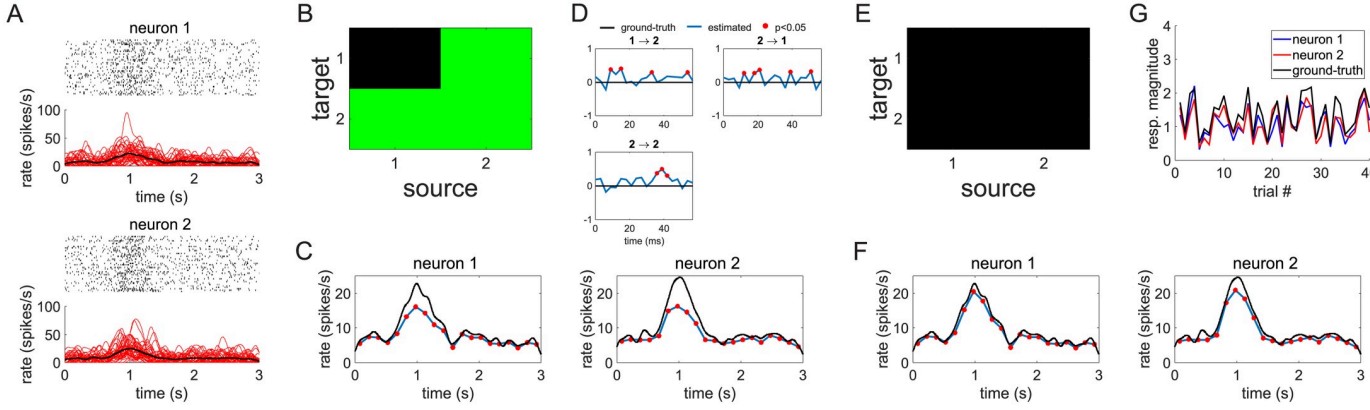

**Fig 8. Trial-by-trial variability and Granger causality.** The panel exemplify how trial-by-trial variability can affect Granger causality measures even when non-stationarity in firing rates is taken into account and how G-ETMV can successfully address this issue. (A) Ground-truth connectivity of our simple 2-neuron system (i.e. no connectivity). (B) Spike trains of the two independent units undergoing non-stationary changes in their firing rates and correlated trial-by-trial variability of response magnitudes. The red curves in the bottom panel represent the firing rate of the single trials, while the thick black line represents the average firing rate. (C) Ground-truth trial-by-trial variability of the responses of the two neurons. The black curve represents, for each trial, the overall level of activation of the two units. (D) In the presence of correlated trial-by-trial variability our G-ETM method recovers an incorrect connectivity pattern. (E-F) Ground-truth values (black curves) and estimates (blue curves) of the exogenous components of the firing rates (panel E) and of the interaction functions for the significant causal connections (panel F) recovered by G-ETM. (G-H) The correct patterns of connectivity (panel G) and exogenous components (panel H) are instead recovered by our G-ETMV method. (H) G-ETMV also provides a faithful estimate of the trial-by-trial response variability of both neurons. This figure is replotted, enlarged, in S2 Fig of the Supplementary Material.

$\beta_{i,p}$ additional parameters that model the overall magnitude of neuron $i$'s response in trial $p$ (Eq 7. See Methods section for further details). Application of this new model (G-ETMV: Granger causality with Exogenous Temporal Modulations and trial-by-trial Variability) to the spike patterns in Fig 8B did not only recover the correct pattern of connectivity (compare Fig 8A and 8G) but it also provided a faithful estimate of the response magnitudes $A_{i,p}$ across trial and neurons (Fig 8C). Furthermore, it also provided a more precise estimate of the exogenous temporal modulations of neuronal responses (Fig 8H). Taken together, results in Fig 8 further support the notion that, in Granger causality, the presence of unaccounted variance (in this case trial-by-trial variability) can produce spurious patterns of functional connectivity.

We next quantitatively compared G-ETM and G-ETMV by means of a series of Monte Carlo simulations. These simulations had the same structure as those in Fig 4 with the notable difference that, to produce correlated trial-by-trial variability the firing rates of all neurons were multiplied, on each trial, by the same factor randomly selected in the interval [.55, 2.05). Consistent with the intuition provided by Fig 8 application of our G-ETM method produced a false positive in 8.5% of the cases; a value that exceeds the set statistical threshold of $p < 0.05$ (Fig 8). On the contrary, G-ETMV not only provided a better estimate of the connectivity patterns (97% vs. 92% correct for the G-ETM and G-ETMV models respectively) but also maintained the percentage of false positives within the set statistical threshold (3.5%, Fig 8). These results show that G-ETMV is an effective technique to estimate causal influences between neurons that exhibit exogenous temporal modulations in their firing rates whose magnitude is variable across trials and correlated across units.

## Discussion

A fundamental goal of Neuroscience is to characterize the brain functional circuits underlying perception, cognition and action. Granger causality addresses this problem by detecting functional influences between simultaneously recorded physiological signals [15]. In previous work, Kim and co-workers proposed a point-process extension of Granger causality that allowed to investigate functional connectivity directly at the spike-train level [7]. As any *standard* Granger causality techniques also Kim et al.'s technique assumes that input time series are jointly stationary. That is, their temporal modulations must be entirely due to the series' past histories. This assumption is however rarely met in real neurophysiological experiments. Indeed, neuronal networks are characterized by a high degree of convergence and the activity of a given neuron is the result of the integration of the outputs of many, potentially thousands, projecting units, which is often not technically possible to concurrently record. Furthermore, brain networks often exhibit slow changes in their global state, which makes the magnitude of neuronal responses vary across trials and be correlated between units.

Here, we first showed that application of standard point-process Granger causality to spike trains that exhibit exogenous temporal modulations produces a non-negligible number of artefactual causal links between neuronal activities. In an experimental setting, these results would suggest the existence of fictitious connectivity patterns and would induce incorrect conclusions concerning the underlying functional influences between neurons. To overcome these problems, we proposed here two novel point-process Granger causality techniques: G-ETM and G-ETMV. G-ETM is computationally less demanding and specifically designed for the case of spike trains exhibiting temporal modulations while G-ETMV is more computationally demanding but also handles the case of trial-by-trial, potentially correlated variability in neuronal responses. The choice of which one to use depends on a trade-off between available computational resources and a-priori hypotheses that the Experimenter has concerning a specific data set.

The jointly stationarity assumption gives Granger causality several appealing characteristics [15]. However, at the same time, it greatly limits its potential applications, as very often we are interested in investigating the functional connectivity of brain networks undergoing stimulus-evoked state transitions whose causes are exogenous to the networks themselves. To extend Granger causality to these cases, two main, not mutually exclusive, methods have been proposed in the literature. The first method consists in performing some form of pre-processing on the data to render them stationary and then apply Granger causality to this *new* stationary data set. For example, simple linear trends can be removed by differentiation while more complex non-stationary components can be removed by subtracting the ensemble average or the estimated evoked response from each trial [16, 17]. These techniques are however designed for time-continuous or continuously sampled data and cannot be directly applied to spike trains given their point-process nature. Furthermore, the removal of the ensemble average assumes that each trial is a realization of the same underlying stochastic process, an assumption that is not always met in practice [13]. The second method consists in using time-varying models to fit the data [18, 19]. These extensions to Granger analysis can effectively deal with time series exhibiting exogenous temporal modulations. However, they possess no underlying test statistics and thus significance of the estimated parameters and model comparison must be assessed by means of empirical and computation-intensive bootstrapping techniques [18, 19].

The Granger causality techniques proposed here overcome both problems. Since they directly model the neurons' CIF, they can be applied to point-process data. Furthermore, they use time- and trial-dependent models of neuronal responses and can thus recover the correct patterns of directed connectivity from spike trains containing exogenous temporal modulations and trial-by-trial variability. Notably, both techniques use generalized linear models to estimate the underlying neuronal CIF. Thus, we could use the rich theoretical framework developed for this class of models and, particularly, the test statistics developed to assess the goodness-of-fit of a given model and the significance of the estimated parameters. This aspect is particularly relevant for Granger causality analysis as this technique is heavily based on model comparison. Finally, both G-ETM and G-ETMV produce an estimate of the effects of both observed and unobserved causes on neuronal responses. Thus, in addition to estimating functional connectivity, they can be also used to decompose the spiking activity of each unit into endogenous (i.e. observed) and exogenous (i.e. unobserved) components.

At the practical level, the results of our Monte Carlo simulations stress the importance of carefully checking that the data set under scrutiny meets the assumptions of Granger causality [20]. Indeed, as shown in Figs 1, 3, 4B, 8D and 9A, applying Granger causality analysis to spike trains that violate the assumptions of a given model produces a number of false positive (i.e. artefactual) functional connections well above the selected significance level. In these cases, incorrect conclusions might be drawn concerning the underlying connectivity pattern.

In addition to clear advantages over Kim et al.'s method, our G-ETM and G-ETMV point process Granger-causality techniques have also intrinsic limitations that need to be discussed. First, both G-ETM and G-ETMV need neuronal activities to be sorted into trials. Therefore, they can be applied only when neuronal data can be meaningfully arranged in this manner. Second, they cannot be applied to cases in which the temporal unfolding of exogenous modulations is not consistent across trials. Point-process methods have been developed that can model some types of trial-by-trial variability in exogenous modulations [21]. However, their inclusion in a Granger causality framework is highly non-trivial and further studies are needed to evaluate its feasibility. Third, G-ETM and G-ETMV detect functional connections by relating patterns of spiking activity within and across units. They might thus fail in cases in which regularities in neurons' firing rates might create spurious relationships between the activities of different units. An, admittedly extreme, example are very regular, but independently

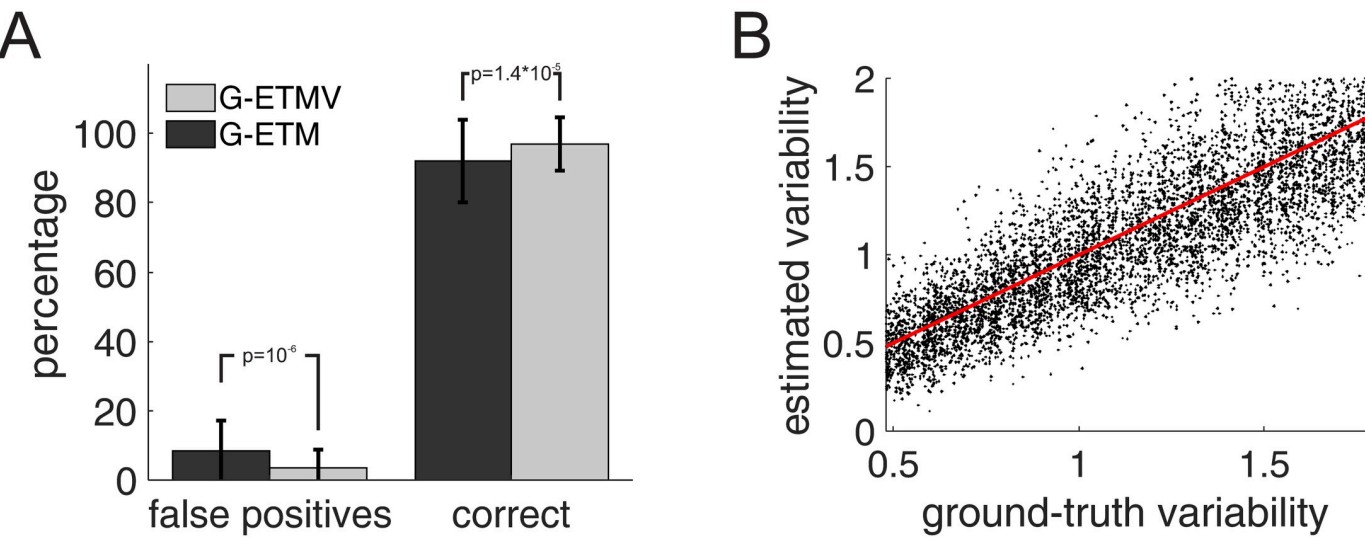

**Fig 9. Monte Carlo comparison of G-ETM and G-ETMV.** (A) Percentage of correct and false positive connections estimated by G-ETM and G-ETMV respectively when applied to spike trains exhibiting both exogenous temporal modulations and trial-by-trial variability. Monte Carlo simulations have the same structure as in Fig 8A. (B) Scatterplot of estimated (i.e. the terms $A_{i,p}$ in Eq 7 vs. ground-truth trial-by-trial variability coefficients. The dots cluster around the unitary slope line (red line), indicating that estimates were close to their ground-truth values.

generated, patterns of spiking activity. Such responses are produced, for example, by networks of non-functionally connected integrate-and-fire units with no synaptic noise and fixed spiking threshold (Fig 10A). The regularities of these firing patterns create synchronizations at a short time-scale between and within units, which might be incorrectly interpreted by our G-ETM method as a pattern of functional connectivity (Fig 10B), with spurious interaction functions (Fig 10C) and exogenous components (Fig 10D). It must be emphasized that spike trains as those shown in Fig 10A are hardly found in physiologically intact preparations. Thus, although of interest from a theoretical standpoint, they have little bearing on the applicability of our G-ETM and G-ETMV methods to physiologically plausible spike trains. Fourth, as is

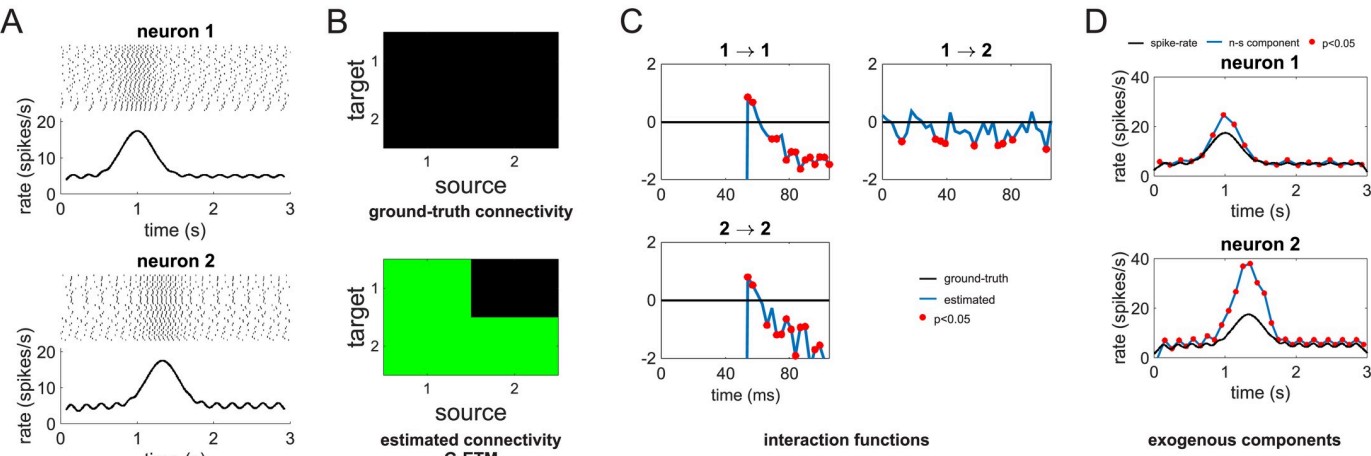

**Fig 10. Application of G-ETM to extremely regular spike trains.** (A) Spike trains generated by a set of 2 integrate-and-fire units with no synaptic noise and fixed spike threshold. (B) Ground-truth (upper panel) and estimated (bottom panel) connectivity. In this case, G-ETM erroneously interpret the spike-level synchrony artificially produced by the regularity of the spiking behaviors as a fictitious pattern of functional connectivity. (C,D) Interaction functions (C) and exogenous components (D) detected by G-ETM. This figure is replotted, enlarged, in S3 Fig of the Supplementary Material.

the case for any model, G-ETM and G-ETMV are based on a set of assumptions. We showed two cases, in which G-ETM is robust (Fig 5) or not (Fig 10) to specific violations of its underlying assumptions respectively. A thorough assessment of its behavior under general conditions of structural uncertainty goes beyond the scope of this work and might need sophisticated statistical methods proposed in the literature (e.g. [22, 23]).

In summary, we presented here two novel point-process Granger analysis techniques, namely G-ETM and G-ETMV, that can correctly detect directed influences between neurons whose responses exhibit exogenous temporal modulations and correlated trial-by-trial variability. These novel techniques allow to investigate the functional connectivity between them during stimulus-evoked responses and thus to reveal how neurons interact not only during baseline conditions, but also when their responses are modulated by exogenous stimulation.

## Methods

We first briefly review the point process Granger causality method proposed by Kim and co-workers [7]. In the following, we will use a lowercase notation for variables that are directly computed from the data and an uppercase notation for variables that are estimated from the GLM fitting process.

A point process is a time series of discrete events that occur in continuous time [24]. Given an observation interval $(0, T]$, let $0 < u^i_1 < \cdots < u^i_j < \cdots < u^i_{J^i} \leq T$ be a set of $J^i$ spike times point process observations for $i = 1, \cdots, Q$ recorded neurons. Let $N_i(t)$ denote the number of spikes of neuron $i$ in the time interval $(0, t]$ with $t \in (0, T]$. A point process model of a spike train is completely characterized by its conditional intensity function (CIF) $\lambda_i$, given the past spiking history $H_i(t)$ of all neurons in the ensemble:

$$\lambda_i(t|H_i(t)) = \lim_{\Delta \to 0} \frac{Pr[N_i(t + \Delta) - N_i(t) = 1|H_i(t)]}{\Delta} \tag{2}$$

where $H_i(t)$ denotes the spiking history of all the neurons in the ensemble up to time $t$ including neuron $i$ itself.

The function $\lambda_i$ needs to be estimated from data. To this end, we first computed the history $H_i(t)$ of each neuron $i$ in $M_i$ non overlapping rectangular windows of duration $W$. We then denoted with $R_{q,m}$ the spike count of neuron $q$ $(1 < q < Q)$ in the interval $m$ $(1 < m < M_i)$ and used a generalized linear model (GLM) framework to model the logarithm of the CIF as a linear combination of the $R_{q,m}$ [25, 26]:

$$log\lambda_i(t|\gamma_i, H_i(t)) = \gamma_{i,0} + \sum_{q=1}^{Q}\sum_{m=1}^{M_i}\gamma_{i,q,m}R_{q,m}(t) \tag{3}$$

where $\gamma_{i,0}$ relate to a baseline level of activity of neuron $i$ and the to-be-estimated interaction function $\gamma_{i,q,m}$ represents the effect of ensemble spiking history $R_{q,m}(t)$ on the firing probability of neuron $i$.

Casting the estimate of $\lambda_i$ into an auto-regressive GLM framework allows an extension of Granger causality to point processes [7]. Indeed, following the definition of Granger causality, one can infer the potential causal connection $j \to i$ of neuron $j$ onto neuron $i$ by comparing the deviance of the full model in Eq 3 with that of a reduced model $\lambda^j_i$ that excludes the effects of neuron $j$ onto neuron $i$:

$$log\lambda^j_i(t|\gamma_i, H_i(t)) = \gamma_{i,0} + \sum_{q=1,q\neq j}^{Q} \sum_{m=1}^{M_i}\gamma_{i,q,m}R_{q,m}(t) \tag{4}$$

If both models describe the data well then the difference of their deviances can be asymptotically described by a chi-square distribution and one can then use the theoretical machinery developed for this distribution to infer statistical significance [7].

## Accounting for temporally modulated spike trains

An assumption of standard Granger causality is that the examined stochastic processes are jointly stationary. That is, their temporal evolution must be entirely due to their past histories. To easily convince ourselves why this is the case, let us look at Eq 3. In this equation, the CIF is assumed to depend, through the terms $R_{q,m}(t)$ only on the past history $H_i(t)$ of the neuronal ensemble. If the statistics of the spike trains are jointly stationary so are also the terms $R_{q,m}(t)$. This ensures that the GLM fitting process will converge to meaningful values for the parameters $\gamma$ and that the difference of the deviances of models 3 and 4 will asymptotically follow a chi-square distribution. However, in the presence of spike trains exhibiting exogenous temporal modulations, the terms $R_{q,m}(t)$ will also be, in general, non-stationary and thus the GLM fitting process may converge to non-meaningful values or not converge at all. Furthermore, the model in Eq 3 will, in general, no longer provide a good description of the data. As a consequence, the deviances of models 3 and 4 might no longer asymptotically follow a chi-square distribution. In this case, the problem of statistically comparing them may even become ill-posed.

To overcome this limitation we first need to understand the characteristics of temporal modulations in spike trains. Neurophysiological experiments are usually organized into trials. Within each trial, an experimental event occurs (e.g. a sensory stimulus is presented, a movement is performed, etc.) that produces modulations in neuronal activities. For data analysis purposes, the continuously recorded neuronal spike trains are then off-line segmented into trials centered around the presented experimental event. A common assumption in analyzing neuronal responses is that the modulations produced by the exogenous event has the same time-course and amplitude across trials. Under this assumption we can thus deal with this non-stationarity by explicitly including it in our model.

To this end, for each neuron $i$ we subdivide the duration $T$ of each trial into $N_i$ non-overlapping windows of duration $T/N_i$. Within each window we then model the CIF as the sum of the to-be-estimated effect of an exogenous event (the experimental event) and the influences of the other neurons. Our model becomes thus:

$$log\lambda_i(t|\gamma_i, H_i(t)) = \alpha_{i,c_i(t)} + \sum_{q=1}^{Q}\sum_{m=1}^{M_i}\gamma_{i,q,m}R_{q,m}(t) \qquad (5)$$

where $0 < t < T$ and $c_i(t) = \lceil \frac{t}{T}N_i \rceil \in \mathbb{N}$ and $1 < c_i(t) < N_i$, is a piece-wise constant integer function indexing a set of $N_i$ additional parameters (one for each of the intervals in which we have subdivided a trial for neuron $i$) that explicitly model changes in firing rates due to exogenous effects (i.e. effects not due to interactions with self or other neurons).

Model parameters were estimated by means of a GLM fitting process with a binomial distribution function and a logit link function. The potential causal influence of neuron $j$ onto neuron $i$ is assessed, similar to the method proposed by Kim et al. [7], by comparing the deviance of the model in Eq 5 with that of a reduced model $\lambda_i^j$ that excludes the effects of neuron $j$ onto

neuron $i$:

$$log\lambda_i^j(t|\gamma_i, H_i(t)) = \alpha_{i,c_i(t)} + \sum_{q=1,q\neq j}^{Q}\sum_{m=1}^{M_i}\gamma_{i,q,m}R_{q,m}(t) \qquad (6)$$

Notably, the GLM fitting process provides not only an estimate of the interaction functions $\gamma_{i,q}$ but also of the exogenous modulations $\alpha_{i,c}$ of neuronal responses. To set the values of the hyper-parameters $M_i$ and $N_i$ we repeated the fitting process using models having different values of $M_i$ and $N_i$ and we then selected the model that minimized Akaike's information criterion (AIC) [7, 27].

## Accounting for trial-by-trial variability

We have so far assumed that stimulus-evoked responses are stereotyped and that their trial-by-trial variability is entirely due to a noise process. However, neuronal responses can exhibit considerable task-related variations across trials that cannot be captured by a noise process. Notably, correlated variations of response magnitudes can modulate cross-correlation or spectral coherence measures resulting in spurious patterns of Granger causality [13, 16]. To avoid these artifacts we need to explicitly include in our model potential trial-by-trial variations in response magnitudes. To this end, we added to our model a set of parameters $\beta_{i,p}$ that represents the amplitude of the non-stationary response component of neuron $i$ in trial $p$:

$$log\lambda_{i,p}(t|\gamma_i, H_i(t)) = \beta_{i,p} + \alpha_{i,c_i(t)} + \sum_{q=1}^{Q}\sum_{m=1}^{M_i}\gamma_{i,q,m}R_{q,m}(t) \qquad (7)$$

where $\lambda_{i,p}$ is the CIF of neuron $i$ in trial $p$. Notably, the fitting process produces also an estimate of the parameters $\beta_{i,p}$ whose values can be used to assess the consistency of response magnitudes across trials. Also in this case, the potential causal influence of neuron $j$ onto neuron $i$ is assessed by comparing the deviance, across all trials, of the model in Eq 7 with that of a reduced model $\lambda_{i,p}^j$ that excludes the effects of neuron $j$ onto neuron $i$. At the implementation level, additional constraints had to be added to ensure convergence in the GLM fitting process of Eq 7. Indeed, if $\bar{\beta}$ and $\bar{\alpha}$ are solutions of Eq 7 then so are $\bar{\beta} - a$ and $\bar{\alpha} + a$, with $a \in \mathbb{R}$. Under these conditions the fitting process would not converge as it would get stuck in a "runaway" process in which $a$, and consequently $\beta$, are indefinitely increased or decreased. We thus added, to our set of predictors, two regularizers that avoid indefinite increase or decrease of the parameters $\beta$.

## Generation of synthetic spike trains

For our simulations we set the temporal granularity to 1 ms. For each neuron $i$ and trial $p$, spike trains were then generated by extracting, for each trial and 1 ms interval, a random number $r$ uniformly distributed between 0 and 1. A spike was assumed to have occurred if $r \leq \lambda_{i,p}(t|\gamma_i, H_i(t))\Delta$ (where $\lambda_i$ represents the time-dependent firing rate in spikes per second and $\Delta = 0.001\ s = 1\ ms$); otherwise, no spike was generated.

At each time $t$, the firing rate $\lambda_{i,p}$ was computed as:

$$\lambda_{i,p}(t) = A_{i,p} \cdot (\lambda_{i,p}^0 + B_i e^{\frac{(t-\tau_i)^2}{\tau_0}}) \cdot \sum_{q=1}^{Q}\sum_{m=1}^{M_i}\delta_{i,q,m}R_{q,m}(t) \qquad (8)$$

where $A_{i,p}$ models trial-to-trial variations of the activity of neuron $i$, $\lambda_{i,p}^0$ is a baseline level of

activity, $B_i e^{\frac{(t-\tau_i)^2}{\tau_0}}$ is a non-stationary Gaussian-shaped modulation of the spike rate centered, within each trial, at time $\tau_i$ and with $\tau_0$ determining its duration. The term $\sum\sum\ldots$ represents the influence of all other neurons including neuron *i* itself. The network topology as well as the functional interactions between neurons are determined by appropriately setting the parameters $\delta_{i,q,m}$. In all our simulations we set $\tau_0 = 200$ *ms*.

In a separate set of simulations (Fig 5) spike trains were generated by means of a network of 4 integrate-and-fire units. This network was implemented in Matlab, from code publicly available on the Matworks website (https://www.mathworks.com/matlabcentral/fileexchange/ 50339-easily-simulate-a-customizable-network-of-spiking-leaky-integrate-and-fire-neurons), which we changed in two ways. First, we modified the equation of the membrane potential so as to simulate, for simplicity, integrate-and-fire rather than leaky integrate-and-fire units. Second, in between spikes, we reset the spike generation threshold according to an exponential distribution so as to generate Poisson spike rates. This assumption is motivated by experimental studies showing that neuronal responses in many cortical areas follow a Poisson distribution (see, for example, [28]). A random voltage threshold can be shown to be equivalent to the physiologically inevitable random noise present in neuronal input currents [11]. In simulations shown in Fig 10, after each spike, we set instead the spike generation threshold of the integrate-and-fire units to a constant value so as to generate perfectly regular spiking patterns.

## Supporting information

**S1 Fig. Spike trains recorded from the monkey pre-motor cortex (area F5) and included in our analysis.** For each neuron, the upper panel shows the spike trains. Here, each row represents a trial and vertical lines mark the occurrences of spikes. The curve in the bottom panel represents the average, across all trials, of the neuron's firing rate.
(EPS)

**S2 Fig. Replot of Fig 8 in the main text: Trial-by-trial variability and Granger causality.** The panel exemplify how trial-by-trial variability can affect Granger causality measures even when non-stationarity in firing rates is taken into account and how G-ETMV can successfully address this issue. (A) Ground-truth connectivity of our simple 2-neuron system (i.e. no connectivity). (B) Spike trains of the two independent units undergoing non-stationary changes in their firing rates and correlated trial-by-trial variability of response magnitudes. The red curves in the bottom panel represent the firing rate of the single trials, while the thick black line represents the average firing rate. (C) Ground-truth trial-by-trial variability of the responses of the two neurons. The black curve represents, for each trial, the overall level of activation of the two units. (D) In the presence of correlated trial-by-trial variability our G-ETM method recovers an incorrect connectivity pattern. (E-F) Ground-truth values (black curves) and estimates (blue curves) of the exogenous components of the firing rates (panel E) and of the interaction functions for the significant causal connections (panel F) recovered by G-ETM. (G-H) The correct patterns of connectivity (panel G) and exogenous components (panel H) are instead recovered by our G-ETMV method. (H) G-ETMV also provides a faithful estimate of the trial-by-trial response variability of both neurons.
(EPS)

**S3 Fig. Replot of Fig 10 in the main text: Application of G-ETM to extremely regular spike trains.** (A) Spike trains generated by a set of 2 integrate-and-fire units with no synaptic noise and fixed spike threshold. (B) Ground-truth (upper panel) and estimated (bottom panel) connectivity. In this case, G-ETM mistakes the spike-level synchrony artificially produced by the regularity of the spiking behaviors as a fictitious pattern of connectivity. (C,D) Interaction

functions (C) and exogenous components (D) incorrectly detected by G-ETM.
(EPS)

## Acknowledgments

We would like to thank John Assad for many stimulating discussions and Daniel Chicharro for providing enlightening comments on a preliminary version of this manuscript.

## Author Contributions

**Conceptualization:** Antonino Casile, Rose T. Faghih, Emery N. Brown.

**Funding acquisition:** Antonino Casile, Emery N. Brown.

**Investigation:** Antonino Casile.

**Methodology:** Antonino Casile.

**Writing – original draft:** Antonino Casile.

**Writing – review & editing:** Antonino Casile, Rose T. Faghih, Emery N. Brown.

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
