## [Decision Letter · Decision Letter 0]

24 Mar 2020

Dear Dr. Casile,

Thank you very much for submitting your manuscript "Robust point-process Granger causality analysis in presence of exogenous temporal modulations and trial-by-trial variability in spike trains." for consideration at PLOS Computational Biology.

As with all papers reviewed by the journal, your manuscript was reviewed by members of the editorial board and by several independent reviewers. In light of the reviews (below this email), we would like to invite the resubmission of a significantly-revised version that takes into account the reviewers' comments.

The manuscript is of considerable technical complexity. In order to make this comprehensible for the general readership of PLOS CB, the arguments in the manuscript need to be clarified. The reviewers have numerous constructive suggestions for this, and also for additional simulations that will help you to better support your conclusions.

We cannot make any decision about publication until we have seen the revised manuscript and your response to the reviewers' comments. Your revised manuscript is also likely to be sent to reviewers for further evaluation.

Sincerely,

Abigail Morrison

Associate Editor

PLOS Computational Biology

Kim Blackwell

Deputy Editor

PLOS Computational Biology

Reviewer's Responses to Questions

**Comments to the Authors:**

Reviewer #1: Review enclosed in pdf format.

Reviewer #2: Overview: The authors extend the existing Granger causality technique of Kim et. al. for continuous time signals to that for discrete spike trains that is robust to within-trial temporal variations in spike and between trial variability in spike magnitudes. The proposed model includes the aforesaid modulations as new parameters into a generalized linear model of the neuronal conditional intensity function. The hyperparameters of this model are tuned by minimizing Akaike’s information criterion. The authors claim that the model shows improvement over the existing Granger causality technique by recovering connectivity matrices with reasonable measures of accuracy and the technique allows decomposition of temporal variations of firing patterns into those effected by endogenous and exogenous components.

Premise: The existing Granger causality technique assumes that all sources of temporal modulations are endogenous to the set of processes considered. The authors herein attempt to highlight that temporal variations may be brought about by both exogenous and endogenous factors via modeling techniques, namely G-ETM. Furthermore, they claim that the computationally involved G-ETMV technique reveals patterns of functional connectivity which is different from synaptic connection motifs but important from a perspective of interpretation.

Main:

• The writeup for the results highlight that considering the effect of exogenous events in each non-overlapping window during each trial predicts the functional connectivity between neural units. However, none of the equations referred to (not even the ones proposed) appear in the main text, rather they all are provided under Methods. As a reader this leads to confusion: how to interpret the proposed model from the text without visualizing the actual mathematical change proposed to the existing technique.

• Building on the above: it would be *very* useful to have some sort of schematic showing a clear comparison between the proposed and prior modeling frameworks. Seeing the core equations side by side, e.g., in a Table, would allow the reader to clearly delineate the mathematical changes and understand the conceptual contribution of the proposed method.

• Figure 2 includes caption for subfigure C, i.e., estimates of the exogenous components of firing patterns, the subfigure itself is not shown in the current version.

• The authors claim that the simulations leading to Figure 3 depict the scenario when simultaneous recording from two different areas is done during an experiment. The idea is to investigate the nature of functional connectivity between subpopulations during such experimental setup. The ground truth model however did not have any interpopulation interactions and what the simulation suggests is that the introduced method could identify that there were no connections between subpopulations. However, it fails to capture how this method(G-ETM) would perform if there were weak/strong interactions between the two subpopulations. Intuitively this should be explored as well. Alternatively, if there are any reasons for omitting such connectivity in model simulations that should be stated.

• The point about self and mutual interactions being inhibitory/excitatory over different timescales needs further clarification/explanation.

• The discussion section does not include any discussion about the limitations/ potential sources of error while using these models that is if there are certain situations when the functional connectivity between neural units are not faithfully recovered using this method. In particular, the proposed method chooses a specific functional form for the new exogenous covariates. Are there situations where these parametric forms are expected to fail? Simulating spike trains from a mismatched model, perhaps even a spiking neuronal network model where a variety of modulating factors could be investigated, could be quite useful in this regard.

Other issues:

• In the Abtract/Intro there is some confusion regarding what is meant by ‘exogenous’. Spike rate modulation may not actually be exogenous to the population; rather it is exogenous to the prior granger formulation. Would it be better to simply refer to these as unmodeled factors?

• The Introduction could use a little more in terms of background on prior Granger methods. The authors introduce the abbreviations for their proposed techniques without actually defining them; it would be useful to get some initial context on what these methods are going to do (e.g., by adding new covariates to the point process framework).

• The notation is Equation (4) is somewhat nonstandard. I think this would be easier to follow if c was defined as an integer and used in the equation, with the (t/T)N_i definition being introduced later. Why are some parameters notated with uppercase while others lowercase?

**Have all data underlying the figures and results presented in the manuscript been provided?**

Reviewer #1: No:

Reviewer #2: No: Primary data are not provided.

PLOS authors have the option to publish the peer review history of their article (what does this mean?). If published, this will include your full peer review and any attached files.

Reviewer #1: Yes: P.G.L. Porta Mana

Reviewer #2: No
---

## [Decision Letter · Decision Letter 1]

31 Aug 2020

Dear Dr. Casile,

Thank you very much for submitting your manuscript "Robust point-process Granger causality analysis in presence of exogenous temporal modulations and trial-by-trial variability in spike trains." for consideration at PLOS Computational Biology. As with all papers reviewed by the journal, your manuscript was reviewed by members of the editorial board and by several independent reviewers. The reviewers appreciated the attention to an important topic. Based on the reviews, we are likely to accept this manuscript for publication, providing that you modify the manuscript according to the review recommendations.

Sincerely,

Abigail Morrison

Associate Editor

PLOS Computational Biology

Kim Blackwell

Deputy Editor

PLOS Computational Biology

[LINK]

Reviewer's Responses to Questions

**Comments to the Authors:**

Reviewer #1: Please see enclosed PDF file.

Reviewer #2: The authors have carefully addressed my prior comments. I have no further concerns.

**Have all data underlying the figures and results presented in the manuscript been provided?**

Reviewer #1: **No: **The authors state they will provide some of the scripts upon acceptance

Reviewer #2: Yes

PLOS authors have the option to publish the peer review history of their article (what does this mean?). If published, this will include your full peer review and any attached files.

Reviewer #1: **Yes: **PGL Porta Mana

Reviewer #2: No
---

## [Decision Letter · Decision Letter 2]

17 Nov 2020

Dear Dr. Casile,

We are pleased to inform you that your manuscript 'Robust point-process Granger causality analysis in presence of exogenous temporal modulations and trial-by-trial variability in spike trains.' has been provisionally accepted for publication in PLOS Computational Biology.

Best regards,

Abigail Morrison

Associate Editor

PLOS Computational Biology

Kim Blackwell

Deputy Editor

PLOS Computational Biology

Reviewer's Responses to Questions

**Comments to the Authors:**

Reviewer #1: I thank the authors for addressing all my concerns.

**Have all data underlying the figures and results presented in the manuscript been provided?**

Reviewer #1: **No: **Authors say data will be provided upon acceptance

PLOS authors have the option to publish the peer review history of their article (what does this mean?). If published, this will include your full peer review and any attached files.

Reviewer #1: **Yes: **PGL Porta Mana

---

## [Editor Report · Acceptance letter]

21 Jan 2021

PCOMPBIOL-D-20-00115R2 

Robust point-process Granger causality analysis in presence of exogenous temporal modulations and trial-by-trial variability in spike trains.

Dear Dr Casile,

I am pleased to inform you that your manuscript has been formally accepted for publication in PLOS Computational Biology. Your manuscript is now with our production department and you will be notified of the publication date in due course.

With kind regards,

Alice Ellingham
